# Exploring the motivations of female community health volunteers in primary healthcare provision in rural Nepal: A qualitative study

**Sarita Panday** [1]*, **Edwin van Teijlingen**[2], **Amy Barnes** [3]

**1** School of Health and Social Care, University of Essex, Colchester, United Kingdom, **2** Faculty of Health and Social Sciences, Bournemouth University, Bournemouth, United Kingdom, **3** Department of Health Sciences, University of York, York, United Kingdom

* s.panday@essex.ac.uk

## Abstract

Motivating Community Health Workers (CHWs)—many of whom are volunteers—is crucial for achieving Universal Healthcare Coverage (UHC) for Primary Healthcare (PHC) in resource-poor areas. In rural Nepal, PHC is mostly delivered by female CHWs, locally known as Female Community Health Volunteers (FCHVs), but little is known about them. This paper explores experiential factors influencing FCHVs' motivations, including how motivation intersects with women's livelihoods and consider what this means for achieving PHC in Nepal and globally. We conducted qualitative research in the hill and the Terai (flatland bordering India) areas of Nepal. Data were purposively collected through 31 semi-structured interviews (20 volunteers, 11 paid local health workers) and three focus group discussions with additional 15 volunteers. All interviews were audio-recorded, transcribed verbatim in Nepali and translated into English. Data were coded using NVivo10, analysed thematically at individual, organisational and community levels. FCHVs' motivations to volunteer was affected in several ways. At the individual level, participants wanted and were committed to voluntary work, yet the opportunity costs of volunteering, out-of-pocket expenditure and inadequate family support strained many of the women who were already overburdened. At the community level, perceived lack of appreciation of volunteer efforts by community members, who saw volunteers as paid health workers, undermined FCHVs motivation to volunteer. Finally, at the organizational level, a bureaucratic emphasis on recording and reporting, and lack of respect from local health workers undermined their motivation at work. Our paper illustrates how FCHVs from some of the poorest backgrounds can be highly motivated to volunteer, yet inadequate social and economic support across individual, organisational and community levels undermined this motivation, the security of their livelihoods, and thus wider efforts to achieve PHC. Financial investments are needed to compensate FCHVs, so that they remain motivated to deliver global health goals for PHC.

**Data Availability Statement:** We conducted interviews and focus group discussions, and this constitutes the data of this manuscript which are not eligible to be publicly shared due to ethical

considerations. The ethical body approving this study is an institutional review board (IRB) of the Nepal Health Research Council (approval letter provided). The contact information for Ethics Committee to which data requests may be sent from interested researchers is approval@nhrc.org.np.

**Funding:** SP received a travel grant from the University of Sheffield to cover the cost of local transportation for data collection within Nepal. The funder had no role in study design, data collection and analysis, decision to publish, or preparation of the manuscript.

**Competing interests:** The authors have declared that no competing interests exist.

## Introduction

Community Health Workers (CHWs) have been key resources in delivering and expanding access to Primary Healthcare (PHC) services, particularly in resource-poor areas of low- and middle-income countries (LMICs) [1–3]. With a shortage of professional health workers, a number of countries in Africa, Asia, and Latin America have used CHWs at a national level to meet the health needs of their population [4]. CHWs are not formally trained or recognised as 'healthcare professionals' but have been trained to promote health in their own communities [5]. They are often the first point of contact between community members and healthcare providers, linking communities to the health system. CHWs are also often seen to be key to reducing health disparities and achieving Sustainable Development Goal (SDG) 3, which aims to ensure healthy lives and promote well-being for all, at all ages (including meeting Target 3.8: achieving universal health coverage) (UHC) [2, 6, 7]. While CHWs can be paid, in many resource-poor health systems, there is a reliance on the unpaid labour of volunteers, often women from resource-poor areas, to carry out this role. However, using women volunteer CHWs to achieve these global health goals without considering their everyday experiences has posed substantial burdens on them [8–11].

Globally, the majority of unpaid CHWs tend to be women who work in poorer parts of Africa and South Asia [2, 12]: for example, the Women's Development Army in Ethiopia [10], Swastha Sewika in Bangladesh [13], and Accredited Social Health Activist in India [14]. Women who are CHWs face obstacles at work that are not faced by their male colleagues, for example, conflict between domestic and work responsibilities and a lack of employment opportunities, yet their socio-economic situation is seldom recognised in CHW programmes [8, 9, 11, 15, 16]. Thus, the reliance on unpaid volunteers has resulted in mainly women taking on this role and, due to the multiple social and community roles women are doing, can undermine the CHWs' motivation in resource-poor areas of LMICs [17]. Motivated CHWs can enhance access to health services and promote people's trust, demand and use of such services, thereby helping to realise UHC [6, 8, 18, 19]. In addition, motivated CHWs are likely to perform better and have a higher retention rate than their counterparts [20, 21].

While motivation of health workers can be assessed in different ways, the conceptual framework developed by Franco et al. [22] is particularly useful as it uses a broad and encompassing definition of motivation as: "...an individual's degree of willingness to exert and maintain an effort towards organizational goals." [22] (p. 1255). The framework has been widely used to understand health workers' motivation [23, 24], including CHWs [17], and considers three key determinants: individual-level factors, organisational-level (work context) factors and community-level factors [22]. Individual level factors include: health workers' personal goals, expectations and experiences. Organisational level factors include organisational structures, resources, processes and culture, including organisational feedback about performance, and community-level factors include how health workers' motivation is influenced by: their interaction with community members, community expectations for how services should be provided, and formal and informal feedback.

At the individual level, CHWs can be motivated by various reasons such as, desire to gain a good status in their community, or desire to achieve something worthwhile (altruism) [1, 25]. They can also be motivated due to their perceived self-empowerment from volunteering [25], or with hope to make some incentives associated with their activities, so that they could escape poverty and materially support families and communities [13, 26]. At the organisational level, CHWs are likely to be motivated by being respected as a member of the health system with a clear set of responsibilities [27]. Similarly, they can be motivated due to availability or anticipation of monetary and non-monetary incentives, further employment opportunities, training

and supervision [20]. At the community level, their motivation is influenced by their working and social relationships with other CHWs, and their community [27, 28]. Particularly, CHWs are motivated by recognition and respect from community members [19].

In this article, we use the Franco framework to focus on motivation of CHWs from rural Nepal, analysing their motivation at each level. In Nepal, the main CHWs are known as Female Community Health Volunteers (FCHVs), a role that is exclusively for women. The FCHV programme began in 1988 and the FCHV strategy was subsequently revised three times in 1990, 1992 and 2003, before the formulation of a new FCHV strategy in 2010 [29–31]. When the programme began, FCHVs were provided with 100 Nepalese rupees (NRs) (USD 0.75) per month and a training allowance of NRs 250 (USD1.87). However, monetary incentives were withdrawn in 1990. In 1992, the training allowance was reinstated, but no other incentives for volunteers until 2010, when the new FCHV strategy (2010) emphasised provision of several non-monetary incentives, such as a celebration of FCHV day and provision of uniform [30]. A FCHV fund (NRs 50, 000 (USD 374.25)) was also created at a village level to support volunteers' livelihoods through a loan [29], but a survey of volunteers reported that only 66% of FCHVs had heard of the fund and only 51% were members of the fund [31]. The FCHV strategy (2010) was revised again in 2019 to include incentives for FCHVs to attend work-related meetings according to government provision, included provision of a letter of honour and a sum of NRs 10, 000 (USD74.85) for retiring FCHVs above 60 years old [29].

As of 2023, more than 50,000 FCHVs form a critical human resource for both government and non-government agencies delivering PHC across the country [32]. Their contribution in reducing child mortality and improving maternal health has been internationally recognised [24]. In terms of motivation, volunteers appear interested in taking on new or additional health roles (e.g. measuring blood pressure [33] and committed to their roles, as evidenced by a low attrition rate (<5%) [34]. Yet in-depth empirical studies exploring FCHVs' motivation to deliver PHC services in Nepal are sparse. Glenton et al. [35] highlighted that FCHVs were motivated to volunteer due to social recognition of their services and that paying them was seen as unnecessary or inappropriate in their socio-cultural context. However, these findings mostly represented policymakers' and managers' views of the FCHV programme—male, salaried, public servants—thereby ignoring potential gender bias inherent in volunteering only by women of lower perceived social status [36]. While this research usefully illustrates the views of key decision-makers, the omission of FCHVs views is significant given gendered and socio-economic hierarchies that exist. In consequence, little is known about how women who volunteer—who tend to be some of the most impoverished women living and working in rural areas—perceive volunteering [17], including what they think about not being paid, or whether they feel it is 'fair'. This gap is not unique to Nepal and has been highlighted in other health systems. This limits our understanding of the role of FCHVs in supporting attainment of SDG3 UHC in Nepal by 2030 and also has an impact on others, such as SDG5 gender equity and SDG8 decent work.

It is necessary to fill this knowledge gap by exploring some of the experiential factors that can influence volunteers' motivation, including how motivation is affected by and fits in with women's livelihoods in rural Nepal from women's own perspectives. Listening to volunteers is key, so that programmes and policy to expand PHC can address the everyday realities of implementation of such programmes 'on the ground'. We therefore report on research with and data from FCHVs themselves and their local supervisors. While it has been almost 9 years since we collected our data, there has been very little research on this topic in the intervening period. For example, we found only two studies that reported FCHVs were motivated to work to prevent a growing burden of noncommunicable diseases despite their workload: they were willing to conduct screening tests to detect hypertension to control blood pressure [37] and to

detect cardiovascular disease [38]. However, the first study was a cross sectional survey [37] and did not consider women volunteers' views about how that added task would influence their work motivation. The second study [38] was conducted with 10 volunteers who wanted to participate and were trained on the topic of cardiovascular disease. The perception and experiences of FCHVs can be different to those volunteers working for government funded public health centres, who often receive limited training (18 days of basic training on various PHC topics) [32]. Recent research in Nepal reveals that FCHVs experience issues such as workload and payment [25, 39, 40]. These are issues that we go on to highlight in this current paper, which therefore suggests the continued salience of our previous research [41].

## FCHVs in Nepal

Before describing the methods, here we provide more information about the context in which FCHVs work in Nepal. FCHVs are typically married women of reproductive age (15–49 years), who work in each ward, which is the smallest administrative division of a village/municipality. Their role is to improve PHC mainly maternal and child health through health education, referral and treatment services, and they have a crucial role in supporting people who are the least able to afford health care [42]. FCHVs' availability, their familiarity with the local context (including language) and their ability to recognise health problems and refer people in a timely fashion, are all strengths that differentiate them from other health-care professionals [25, 43]. However, FCHVs in rural Nepal work in precarious conditions.

In Nepal, the socio-economic status of women in rural areas is particularly low and it is in rural areas where the vast majority of FCHVs work (46,088, compared to only a small number of volunteers 5,328 in urban areas as of 2022) [44]. Women spend much of their time looking after their families, which generates no income for them (i.e. it is unpaid care work) and leaves them economically insecure. For example, most women (66.5%) within the labour force work informally and one-third (31.8%) rely on subsistence farming compared to 13.1% men [45]. As indicated above, since 1990, FCHV policy has provided limited financial remuneration for women volunteers and therefore it is in this context that FCHVs continue to carry out their health system roles in a largely voluntary capacity to address both health and non-health issues, as mentioned above [33, 46].

## Methods

### Study design

This study aimed to explore experiential factors that can influence volunteers' motivation, including how motivation is affected by and fits in with women's livelihoods in rural Nepal and from women's own perspectives. Qualitative data was collected in 2014 as part of a wider study [47]. Data was purposively collected through 31 semi-structured interviews (20 volunteers and 11 paid local health workers) and 3 focus group discussions with additional 15 volunteers. Use of interviews and groups discussions enabled us to gather data from individual health workers as well as groups, thus allowing us to understand differing opinions [48].

### Study settings

Two distinct rural settings were selected: one hill region (Dhading District) and one Terai flatland region, on the south plains bordering India (Sarlahi District). These contrasting areas were chosen based on ease of access to PHC services and on the reported 'success' of the FCHV programme in expanding access to basic PHC services [49]. The study villages in Dhading District are well known for 'success' in implementing the volunteer programme despite the

villages being isolated with relatively limited access to PHC; some places are five hours walk from the nearest health centre [50]. In the Terai region, Sarlahi District was chosen because of its relatively easy access to PHC services–mostly about 30 minutes on foot–and also because the district has ethnically different populations in comparison to the hill region; for example Muslim and Madhesi populations are unique to the Terai, and Chepang is unique to the hill village. Focusing on Sarlahi therefore offered a contrasting socio-demographic situation to Dhading.

## Ethics statement

Ethical approval for the study was received from the Nepal Health Research Council Ethical Review Board in 2013 (Registration number 32/2013). Informed written consent was obtained from all the participants included in the study and responses were anonymised. The first author obtained consent to publish from the participants. This was written consent, although for illiterate participants the first author read out the consent form and obtained their written signature (some participants could not read the form but were able and content to sign their names).

Additional information regarding the ethical, cultural, and scientific considerations specific to inclusivity in global research is included in the S1 Checklist.

## Data collection/ study participants

As indicated above, while it is 9 years since the data for this study was first collected, the context in which the Principal Investigator and first author of this manuscript first collected the data is largely unchanged: the numbers of volunteer women doing FCHV roles is similar now to when the data was collected. For example, in 2022, there were 46,088 FCHVs in rural areas compared to only a small number of 5,328 FCHVs in urban areas [32] and there were a similar number in 2014: 47,328 in rural areas compared to 4,142 in urban regions in 2014 [49]. While volunteers' roles and responsibilities were widened in promoting health of people in rural areas to meet the global goals PHC for all, SDG for health (for example, taking on screening non-communicable diseases, providing health education and referring people for health checks [33, 46], as described above) FCHVs remain as unpaid workers.

Data was collected between 9-05-2014 to 21-09-2014. Study participants were recruited and purposefully selected to capture diverse experiences in the two study areas. Selection criteria for FCHVs within the two study areas included ethnicity and duration of volunteering (Table 1). In Dhading, some participants belonged to indigenous groups which are categorised as either highly marginalised (Chepang) or marginalised (Tamang, Bhujel). In the Terai, the study participants comprised both upper caste (Bramhan, Kshatri) and marginalised groups (Madhesi and Muslim) (Tables 1 and 2).

The PI conducted interviews and focus group discussions in Nepali, and it ranged in duration from 15 to 60 minutes. All people approached to take part agreed to participate. Interviews and focus groups were conducted using a semi-structured thematic interview guide (S1 File) and most were conducted in participants' homes, although some individual interviews and group discussions were held in a meeting room at health centres, or local cafes as per participants' availability and preference. For example, the first author conducted a focus group discussion and also conducted an individual interview with a volunteer from a remote area after their monthly meeting at the local health centre. Interviews/focus groups were either audio-recorded or concurrent notes were taken, depending on the consent of the participants and the desirability of keeping conversations unrecorded.

In terms of research-related payments, 15 FCHVs who had to commute to participate in interviews were paid NRs 500 (US $3.74) to cover their expenses. No other incentives were

**Table 1. Socio-demographic characteristics of female community health volunteers (FCHVs).**

| Respondents | Place | Type of Data | Age | Caste/ ethnicity | Education (in years) | Work Experience (years) | Walking distance to health centres |
|---|---|---|---|---|---|---|---|
| 1 | Dhading | Interview | 45–59 | Brahmin | Literate | 15 | 1hr |
| 2 | | | ≥60 | Brahmin | Literate | 15 | 2hrs |
| 3 | | | 45–59 | Brahmin | Literate | 15 | 20 min |
| 4 | | | ≥60 | Tamang | 0 | 15 | 5-6hrs |
| 5 | | | 45–59 | Brahmin | 2 | 16 | 1hr |
| 6 | | | 45–59 | Chhetri | Literate | 24 | 2 min |
| 7 | | | 45–59 | Bhujel | Literate | 24 | 30 min |
| 8 | | | 45–59 | Brahmin | 10 | 7 | 15 min by bus |
| 9 | Sarlahi | | 45–59 | Tamang | 5 | 10 | 30–45 min |
| 10 | | | 45–59 | Tamang | 0 | 10 | 1 hr |
| 11 | | | 45–59 | Tamang | 0 | 19 | 1 hr |
| 12 | | | ≥60 | Tamang | 0 | 25 | 15 min |
| 13 | | | 45–59 | Madhesi | 0 | 19 | 10 min |
| 14 | | | 45–59 | Gurung | Literate | 19 | 25 min |
| 15 | | | 30–44 | Chhetri | 8 | 19 | 1 hr |
| 16 | | | 45–59 | Brahmin | 10 | 19 | 20 min |
| 17 | | | ≤30 | Brahmin | 9 | 3 | 1hr of cycling |
| 18 | | | 45–59 | Lama | 4 | 21 | 25 min |
| 19 | | | 45–59 | Magar | 0 | 19 | 1 hr |
| 20 | | | 45–59 | Madhesi | 10 | 26 | 15 min |
| Focus group 1 | Dhading, Gajuri | | ≤30 | Lama | 10 | 4 | 1.5 hrs |
| | | | ≤30 | Brahmin | 12 | 6 | 30 min |
| | | | 45–59 | Chepang | 0 | 16 | 2 hrs |
| | | | 45–59 | Brahmin | Literate | 15 | 1 hr |
| Focus group 2 | Sarlahi, Harion | | 30–44 | Brahmin | 10 | 19 | 20 min |
| | | | 45–59 | Brahmin | 0 | 19 | 10 min |
| | | | 30–44 | Brahmin | 10 | 19 | 15 min |
| | | | 45–59 | Brahmin | 8 | 19 | 30 min |
| | | | 30–44 | Brahmin | 10 | 19 | 30 min |
| Focus group 3 | Sarlahi- Lalbandi | | ≤30 | Brahmin | 12 | 3 | 20 min |
| | | | 30–44 | Gole | 10 | 7 | 20 min |
| | | | 45–59 | Lama | 10 | 2 | 30 min |
| | | | ≤30 | Brahmin | 10 | 1 | 20min |
| | | | ≤30 | Brahmin | 12 | 1 | 2 hrs |
| | | | 30–44 | Chhetri | 10 | 1 | 10 min |

Literate- Able to read and write Nepali, min- minute, hr-hour

provided to the participants to take part in the study. However, when possible, refreshments (tea, biscuits, snacks, cold drinks) were arranged after the interview/group discussions.

In total, 31 participants were interviewed: 20 FCHVs (Table 1), and 11 paid local health workers (Table 2) who were supervising or working with FCHVs to implement PHC in local villages. Of these 11 paid local health workers, 7 were from public health centres, which was government funded and 4 from local NGOs) (Table 2). In addition, 3 focus groups were conducted with 15 additional FCHVs (Table 1), who were gathered for their monthly meeting at local health centres. One FCHV in the hill region participated in both interview and a group discussion, hence the total number of participants is 45.

**Table 2. Demographic characteristics of paid health workers.**

| Health Workers HW | Place | Position | Working Institution | Caste/Ethnicity |
|---|---|---|---|---|
| 1. | Dhading | Staff Nurse | Government | Brahmin |
| 2. | Dhading | Auxiliary Nurse Midwife (ANM) | Government | Brahmin |
| 3. | Dhading | Auxiliary Health Worker (AHW) | Government | Brahmin |
| 4. | Dhading | District Public Health Officer (DPHO) | Government | Muslim |
| 5. | Dhading | ANM | Non-Government | Indigenous |
| 6. | Dhading | Field Coordinator | Non-Government | Indigenous |
| 7. | Sarlahi | Senior AHW | Government | Madhesi |
| 8. | Sarlahi | Female Community Health Volunteer (FCHV) district supervisor | Government | Madhesi |
| 9. | Sarlahi | AHW | Government | Madhesi |
| 10. | Sarlahi | ANM | Non-Government | Indigenous |
| 11. | Sarlahi | Field Coordinator | Non-Government | Brahmin |

The demographic details of participants are shown on in Tables 1 and 2, which have been adapted from our earlier publication [51].

Table 1 shows FCHVs represented diverse ethnic groups with a majority being Brahmin (n = 17) followed by Tamang and other ethnic groups. Their ages ranged from 25–70 years, with a large minority (15/35) between 45–49 years of age. Many (14/35) did not have formal education, eight were illiterate and another seven could only write their name. Generally, the younger the volunteers, the better education they had received. In terms of work experience, it ranged from 3–26 years with a majority (20/35) between 10–20 year.

Table 2 shows paid health workers represented both government and non-government organisations and were from a diverse range of ethnic groups. The paid local health workers were interviewed from the same localities as the FCHVs.

## Analysis

Data were transcribed verbatim in Nepali and translated into English. In order to increase the study's rigor, multiple coding was applied to part of the data [48]. EvT coded four interviews and a group discussion transcripts in English independently. The codes were compared with those of the PI and any discrepancies were discussed and resolved. The PI then completed the coding using NVivo10 software and analysed the codes using thematic analysis [52]. Data were analysed in an iterative fashion, moving back and forth between transcripts, reflective notes, field notes, and the literature. The reliability of coding and interpretation was also checked during analysis by re-examining the transcripts.

Building on the framework proposed by Franco et al. [22], common themes across the data set were developed in NVivo at three levels. Individual, organisational and community level factors were identified by merging data from different data sources (interviews and focus group discussions/ volunteers and the paid local health workers). Data across volunteers and paid health workers were compared for data triangulation [53]. In addition, at the individual level, subthemes are selected given their recurrence and prominence in our data as shown in Fig 1.

Results are illustrated using quotes from different groups of participants and different methods (interviews and focus groups). Each quote has the following identifiers (age range, years of education; exp = years of experience in the post; and distance = walking distance to health centres in minutes/hrs otherwise stated). The acronym HW represents health workers, and D and S represent the two study areas: Dhading and Sarlahi.

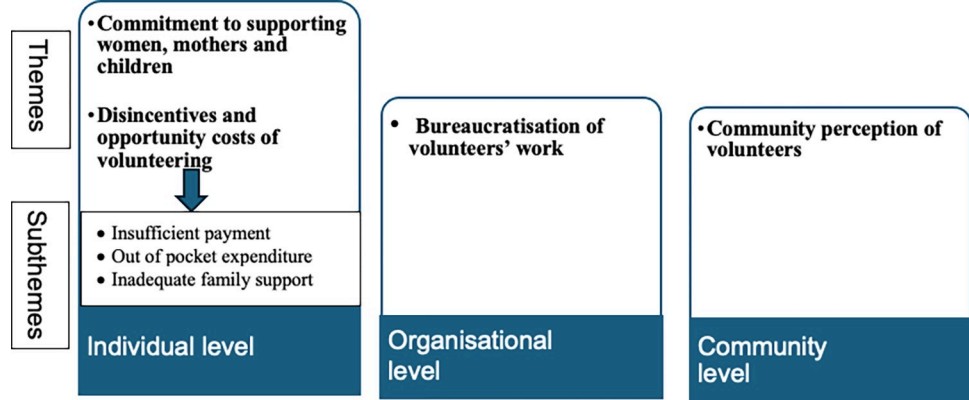

**Fig 1. Illustration of themes and subthemes indicating motivations of female community health volunteers at individual, organisational and community level.**

## Results

The results consider the main factors that influence FCHV motivation across three levels: 1) individual level factors, 2) organizational level factors and 3) community level factors. At the individual level, we present two subthemes: i) FCHVs' commitment to supporting women mothers and children and ii) disincentives and opportunity costs of volunteering. At the organisational level, we present a subtheme 'bureaucratisation of volunteers' work.' Finally at the community level, we present a subtheme 'community perception of volunteers' which covers both community recognition of volunteers and community misperception of volunteers as paid workers. These subthemes are summarised in Fig 1.

### 1. Individual level factors

Individual level factors include: FCHVs' personal goals, expectations and experiences. At the individual level, we found that all volunteers were committed to their unpaid volunteering role. Yet, their experiences of volunteering were different. The time spent in volunteering, high levels of out-of-pocket expenditure and inadequate family support strained women volunteers who were already overburdened, thereby lowering their work motivation at work. However, some volunteers also saw volunteering as an opportunity to earn some money (for example, small amounts in the form of training and travel allowances) or improve their future employment prospects, but with questions as to the extent to which this was achieved in practice. These findings are summarised under two subthemes: i) commitment to supporting women, mothers and children and ii) social and financial disincentives and opportunity costs of volunteering.

**1.1 Commitment to supporting women, mothers and children.** Every volunteer interviewed spoke of being committed to their volunteering work despite its challenges. Many volunteers, especially older illiterate women, spoke about their work delivering health services as a form of basic human and social responsibility; couching their role in terms of serving their own people. Many FCHVs had not had the opportunity to access health information during their own pregnancies and childbirth. However, they reported that they had witnessed a reduction in maternal and child deaths and improvements in women's overall health status in their communities since they became a volunteer. Such changes inspired them to continue volunteering work at personal level as shown in the following quote:

*What I feel good about, my work, is that I am helping pregnant women, mothers and children to save their lives. In the past, sometimes the baby's hands used to come outside the vagina,*

*sometimes their foot used to come outside. It used to cost huge expenses for the families. Now, such incidents have been reduced. We have protected our children and mothers and pregnant women from death, that is the best thing we have achieved. FCHVD2 (≥60, literate, exp 15 years, distance 2hrs).*

Volunteering provided women with opportunities to go outside their houses and meet people, which was a challenge for many women in rural Nepal. Therefore, even older volunteers over 60 years old with physical issues wanted to continue their service, as a volunteer explained:

*I want to walk, I want to talk. Now, what to do? This knee is giving me a little problem. I am not able to see properly. Otherwise, I go everywhere they ask, be it Kathmandu, Delhi or Bombay. I go anywhere. FCHVS12 (≥60, edu 0, exp 25; distance 15min).*

As shown above, at the individual level, volunteers were committed to do their unpaid work, yet all the volunteers in the interviews and group discussions reported personal difficulties they experienced in undertaking their role as a volunteer. These are discussed under sub-theme disincentives and opportunity costs of volunteering.

**1.2 Disincentives and opportunity costs of volunteering.** At the individual level, FCHVs described four key things that undermined their motivation to deliver PHC services: i) insufficient payment, ii) the opportunity costs of volunteering, particularly in terms implications for their livelihoods iii) out of pocket expenditures incurred in volunteering, and iv) inadequate family support to volunteer. First, both volunteers and health workers described a growing set of healthcare responsibilities given to volunteers, for example, reporting health and non-health activities, which were not part of their role when the volunteer programme first began in 1988. Yet, there was little to no financial support available to volunteers in this study. Given these growing responsibilities for volunteers, many referred to the large amount of time they spent in volunteering and the consequent time pressures that this placed on them and on their livelihoods; in other words, they spoke about the opportunity costs of volunteering. Some volunteers in the hill communities reported spending several hours on foot to reach local health centres for monthly training or reporting, which affected their livelihood. As a volunteer from Dhading explained:

*There are not many people to work at home. I have not been able to put paddy seeds for plantation while others [in the village] have already done this. Women in the village don't even have time to go for check-ups, but they want me to stay with them from the morning to the evening during their labour. What should I do? I cannot work for villagers only. I need to look after my animals. I need to eat food, don't I? FCHVD4 ((≥60, Edu 0, exp 15; distance 5-6hrs).*

Both health workers and volunteers agreed that volunteers face a financial disadvantage that discourages or disincentivises volunteering. Given many volunteers did not have other sources of income than subsistence farming, both volunteers and health workers agreed that the incentives volunteers received—NRs 200 (£1.33) to cover their travel cost—was insufficient and did not match with the time spent on volunteering. Volunteers had to pay someone to do their farming work at home, often at higher prices (e.g. NRs 400 (£2.58) per person per day for labour). As a volunteer from Terai explained:

*If we ask someone to work from morning to evening, we need to pay above NRs 400 (£2.58) including their breakfast and lunch. We don't even get that amount for our work. FCHVS6 (45–59, literate, exp 19; distance 25min).*

Some volunteers also described out-of-pocket expenditures that were necessary for them to carry out their health care roles; for example, calling mothers to attend meetings by telephone or calling for an ambulance. Such costs were not reimbursed, which left already poor women volunteers even poorer and undermined motivation. As one FCHV from Dhading commented:

*There is a zero-balance on my mobile due to the mothers' group meeting [shows her mobile]. I need to call the health post to enquire about the whereabouts of the ambulance. This also requires money. . .. Because there is no money, there is not much motivation to work. FCHVD8 (30–44, edu 10, exp 7; distance 15min by bus).*

Consequently, in the absence of payment, some family members thought that volunteers were wasting their time and hence their families—husband, mothers-in-law and children—did not support their volunteering work. For example, a volunteer from Terai commented:

*I was almost forced to stop the volunteering because of the household chores. I felt over-whelmed when I returned home from volunteering. There would be big piles of household chores left for me and without the support of family members, it became almost impossible to continue the volunteering. FCHVS17 (≤30, edu 9, exp 3; distance 1 hour of cycling).*

Yet, despite these challenges we found that FCHVs wanted to continue volunteering because they saw it as an opportunity to advocate their needs and demand what they thought was right and fair. Both unpaid FCHVs and paid health workers indicated that they thought FCHCs should be paid, which would add value to what they were doing. Several volunteers reported that they wanted to be fairly recompensated for their labour via a monetary allowance. As a volunteer from Dhading argued:

*A salary for us would be better. We do all the reporting work. After all the work we have done, we should be paid. FCHVD7 (30–44, literate, exp 24; distance 30min).*

Most volunteers expressed a strong need for monetary compensation, with some describing volunteering (as in the quote above) as 'work.' This was particularly the case for young and educated women volunteers, who saw the volunteering opportunity as a possible entryway into paid work. They were willing to do it without pay for a start, but hoped for a payment, due to the costs of living. As a young volunteer from Sarlahi commented:

*Whether a volunteer should be given a monthly payment, this needs to be considered. It is okay to select those women who can actually work. At this time, it is not possible to work voluntarily. FCHVS15 (30–44, edu 8, exp 19; distance 1 hr).*

Thus, at the individual level, despite the social and financial disincentives of volunteering, we found that women were committed to volunteer. Yet, organisational factors created another challenge for FCHVs.

## 2. Organisational level factors—Bureaucratisation of volunteers' work

A range of organisational factors, including regular training of volunteers, availability of medical supplies, and supported supervision enhanced the motivation of FCHVs to work–with the importance of these factors for volunteers' services reported in our earlier study [25]. The available public health system support in terms of medical supplies and regular supervision of FCHVs were systematically different in the two selected regions (hill villages and the Terai),

which has been reported in our earlier study [25]. However, our data from interviews and focus group discussions showed that excessive bureaucratisation of volunteers' work in the form of recording and reporting to paid health workers undermined their motivation. For example, many volunteers reported being asked by paid health workers (based in local health centres) to complete formal forms of reporting into the health system. This took up a lot of their time, as one FCHV from Dhading commented:

> *There is too much of recording work as compared to the past. Health workers want records of under one-year children, under five-year children, pregnant women, new mothers, family planning cases, the number of pills taken by individuals, women on Depo-Provera injections, intrauterine contraceptive device users, vasectomy cases and so on. We need to calculate the numbers of each case.* FCHVD6 (45–59, literate, exp 24; distance 2 min).

Volunteers also complained that local health workers seemed to emphasise the importance of reporting, rather than the 'actual work' of FCHVs:

> *The health workers never normally ask us how we are working. Only at the time of reporting do they ask us how well we have filled the reports in. The emphasis is on the report, not the work.* Focus group 1 Participant1 (≤30, edu 10, exp 4; distance 1.5 hrs).

A health worker also revealed that some paid health workers were involved in unfair actions towards FCHVs; for example, FCHVs were sometimes provided incentives (an allowance) to cover a single day, instead of the actual intended time for training of three to four days: *The training duration for FCHVs is reduced contrary to the given guidelines, so as to save money from giving allowances to them. If there is training for three or four days then, it would be reduced to one day.* HW11 (Field co-ordinator, government services).

In the absence of adequate financial incentives, many volunteers expressed feelings of being 'fed up' and also gave examples of instances in which they felt undervalued. For example, volunteers indicated how paid health workers often criticised them and/or were disrespectful if they were unable to fill the report card on time. As a volunteer from Dhading indicated:

> *This work does not give credit to us in the village. Health workers ask us, 'did you do this work? If you don't bring the report, who will bring it?' There is no respect, no respect at all in this work.* FCHVD1 (45–59, literate, exp 15; distance 1hr).

Given these issues with how FCHVs felt their emotional and physical labour was undervalued, a small number of FCHVs spoke about how they were trying to mobilise for change. For example, one volunteer spoke about how they wanted to discuss the budget available for volunteers with the local government authority, declaring that she was planning to take collective action against the village secretary:

> *I became a representative as a chairperson of FCHVs from our village and have given them (women) a voice. I have threatened [name], if they would not allocate any budget for us even this year, then I would be bringing the mothers' group to have a word with them.* FCHVS16 (45–59, edu 10, exp 19; distance 20min).

Thus, at the organizational level, we found that the bureaucratisation of volunteers' work, especially emphasis on recording and reporting of health related information, without adequate training and financial support to volunteer, undermined their motivation.

### 3. Community level factors–Community perception of volunteers

At community level, the social experience of working in one's own community was not the same for all volunteers. Some volunteers and health workers reported that the volunteers were motivated to volunteer due to community recognition of their work. For example, a volunteer from Sarlahi highlighted trust of local women towards them:

> *People trust us in the field. We had some difficulties in the past, but now, if FCHVs go there and ask people to eat anything to make them feel better, then they even eat the poison. There will be no doubt on what (medicine) I give her. FCHVS16* (45–59, edu 10, exp 19; distance 20min).

However, our data from interviews and focus groups suggests that volunteers were more concerned about not being valued by local community members than feeling valued or respected. We found a socio-cultural 'clash' between 'the reality' of FCHVs being unpaid volunteers and their 'perceived status' as paid professionals within their communities; not only leading to misunderstandings of what FCHVs do and limited community support, but also undermining volunteers' motivation to work with the PHC system. A typical response is encapsulated by this volunteer:

> *We do not get reward/praise (jas) in the village. Villagers ask us, 'did you work this*? *if you don't bring medicines, who will bring*?' *There is no reward (ausaj), no reward at all in this work. We don't get a salary, but we get blamed for not working well in the local community despite being paid. FCHVD1* (45–59, literate, exp 15; distance 1hr).

FCHVs felt they were being criticized partly because community members perceived them as paid health workers and expected them to spend more hours in volunteering than they were assigned to. This undermined volunteers' motivation at work, as one commented:

> *Some people say, 'she gets a salary every month, but she does not come to our home.' I don't feel like working after listening to this. FCHVD7* (30–44, literate, exp 24; distance 30min).

It appeared that, as more and more functions were given to volunteers, communities' views of volunteers and the social value placed on them changed: seeing FCHVs as 'professional' health workers, instead of volunteers or peers from the community. This reflects an undermining of the relationships that underpin volunteer's motivation to work at the community level.

Thus, our overall our findings show that FCHVs in rural Nepal can be highly motivated to work as a volunteer at the individual level, yet inadequate family, community and organisational (health system) support undermine this, the security of their livelihoods, and thus wider efforts to deliver PHC for all.

## Discussion

This paper has explored motivation of volunteer CHWs from their own perspectives and also included the views of paid local health workers in rural Nepal. Our key findings show that Nepali CHWs–FCHVs–are highly committed and motivated to volunteer but also that the social and financial opportunity costs of volunteering, out-of-pocket expenditures and inadequate family support disincentivize them. In addition, the bureaucratisation of volunteers' work and community misperception of volunteers as paid workers undermined their motivation to volunteer. These key findings are discussed with reference to wider literature and the

current situation in Nepal and are presented at three levels: individual, organizational and community.

At the individual level, we found that despite everyday work challenges, FCHVs were motivated to volunteer for reasons such as altruism, limited monetary incentives and/or social status. They valued the altruism of doing something for the public good and saw volunteering as an opportunity to make a difference to maternal and child health in their communities. While volunteers, regardless of their age, reported that they wanted to be paid for their services, young volunteers emphasised that they joined volunteering expecting a better future and route to financial security or employment. These findings are consistent with CHW literature from sub-Saharan Africa and South Asia [54, 55]. For example, in a study conducted in Bangladesh, India, Kenya, Malawi and Nigeria, involving thirty-two focus group discussions with 361 individuals and 116 key informant interviews with CHWs, health workers and managers, it was found that CHWs consistently expressed a need for appropriate and consistent compensation for their work [54]. Similarly, another study of CHWs that also included their supervisors and high-level officials (n = 95) within Global Polio Eradication Initiative in India, Nepal, Pakistan, Ethiopia and Rwanda showed that when CHWs were provided with some financial compensation, this was perceived to be low and exploitative of CHWs' work [55].

We also found that regardless of a volunteers' age, they wanted to retain their position as a volunteer despite financial challenges, given how they felt it provided women with greater social status and was a route to self-empowerment. For example, in our findings above, and in our previous work on this topic, we highlight how FCHVs appreciated the opportunities to learn new knowledge and skills, and valued travelling outside their houses, meeting new people, and gaining respect from health workers for the work they undertook, which might not be possible for most women in villages [25]. Such empowering experiences can be crucial for women in societies with low social status. Women in Nepal are known to often have low literacy, low socio-economic status and have low prospects of finding paid employment [45], as also seen among CHWs globally and this is a situation that has not changed since the research was conducted [1, 8, 14, 20, 56, 57]. Despite their commitment and individual motivation to take on the FCHV role, we found that women volunteered at considerable social and material cost to themselves and their families, and their expectations were often not fully met in 'the everyday' realities of volunteering. Working without adequate payment threatened volunteers' already precarious living conditions; further overburdening them with their household chores and farming responsibilities. According to the national survey of Nepalese volunteers, average working hours of volunteers increased from 1.7 to 3.1 hours per day between 2006 and 2014 [34], and this appears to continue to be the case. There is however, no more recent data available on volunteer working hours. At the time of the research, and indeed now in Nepal, volunteers have expanded workloads that often involve carrying out work for programmes to detect noncommunicable diseases, such as hypertension (e.g. monitoring blood pressure and educating people about its risk factors) [33, 46]. In line with the current government's policy in Nepal, volunteers continue to be expected to work without compensation. Yet we found that out-of-pocket expenditure incurred in volunteering left already-poor women FCHVs even poorer. The economic insecurities of women CHWs have been reported in other studies from Nepal [25, 39] and in countries including Ethiopia [17], South Africa [58], Bangladesh [59], Pakistan and Sierra Leone [15]. This suggests that women volunteers are consistently mobilised to meet global developmental goals without fully considering their livelihoods and everyday needs, thus undermining their motivation at work and, by extension, access to UHC.

At the organizational level, we found that bureaucratisation of volunteers' work through formal recording and reporting of health activities without financial compensation meant FCHVs saw these requests for their labour as unfair (as evidenced by FCHVs questioning why

they had to volunteer while local health workers were paid for doing similar work) which also undermined their motivation. A situation that was compounded by health workers criticising FCHVs for being unable to produce reports–despite this being unsurprising given that many FCHVs are illiterate. This illustrates however, the subordinate status of volunteers to paid government health workers in Nepal and is a situation that persists to the current day. This situation is not unique to Nepal: women CHWs are often at the bottom of gendered health bureaucracies and experience difficulties in advocating their own needs and those of communities. For example, studies from Ethiopia highlight how women CHWs work without adequate compensation in a structure which reinforces gendered hierarchy [12, 60] and how such hierarchical structures and bureaucratisation demotivate health workers [57].

At the community level, we found that, as more and more functions are given to volunteers to perform over time, communities' views of volunteers and the social value placed on them have changed: seeing FCHVs as 'professional' health workers instead of volunteers from the community and undermining the relationships that underpin volunteer's motivation to work. This change in community perceptions has been driven by the way the FCHV programme has developed in Nepal. When the FCHV programme began in 1988, their roles were limited to provision of health education. However, over the years, they have been involved in the provision of preventive, promotive and curative healthcare activities, including distribution of medicines as part of national campaigns, which is one-off paid activity [61]. It is likely that the expansion of activity, as well as involvement in paid activities by NGOs [40], although rare, has led some community members to view volunteers as paid workers and to be more critical views of volunteers' service shortcomings. This has, however, deprived FCHVs of the community support they need and which underpinned their original motivation to work. Again, these findings are consistent with studies of CHWs from Africa and Asia [15, 54], but contradict other evidence from Nepal [35], which was primarily based on the assumptions of policy-makers and programme managers.

Overall, our findings suggest that financial compensation is crucial for women volunteers—who mostly come from a poor socioeconomic background—to be motivated to deliver global goals for PHC [6, 21, 54, 57, 58]. As highlighted in earlier studies of FCHVs, Nepal needs a system that not only socially recognises volunteers' efforts, but also supports their livelihood: it is only when both these are met that volunteers will be able to deliver their role with the support of family members [46, 62, 63]. This however, has financial implications for achieving health related goals (SDG3), suggesting that more financial investment for FCHVs is needed for integrated action across three levels (individual, organisational and community). For example, motivating volunteers at individual level will require national policy change, not only to re-enact the financial incentives that were present in the FCHV scheme when it first began, but also to ensure that the incentives match their workload and there is a greater recognition of the value of FCHVs in national policy. At organisational level, it will require the implementation of procedures and practices to ensure that FCHVs get all the financial incentives that they are due and in a timely way and also a programme of cultural change to improve the way that paid health workers value and interact with FCHVs to ensure that their relationships become grounded in and engender respect. At community level, integrated action is needed to increase community awareness of FCHVs' roles and their contribution to community health. It is also important to recognise that, in terms of global policy, action in this respect will have knock-on positive effects for other SDGs, including SDG 8, as it will increase women's entry into the workforce, and SDG10, as by strengthening women's social status and livelihoods it will help address gender inequalities [64].

Our study has some limitations. First, data was collected in 2014 from two areas of Nepal using qualitative methods, hence the detailed findings are not generalisable. The thematic

topics identified are however, potentially relevant in other settings in Nepal and other resource-poor health systems, given that they resonate with other existing literature on this topic, as discussed above. Second, participants might have been particularly concerned to express their views on issues of monetary compensation (in line with social desirability bias), using the interviews as a potential chance to highlight their complaints to higher level policy audience and programme managers via the researcher, meaning that other 'everyday issues' may not have been discussed. This does not, however, detract from this being a key issue for the FCHV participants and therefore key issue to be considered and addressed in further research and in policy and practice.

Further research is needed to assess and measure the degree to which the identified factors influence FCHVs' motivation to deliver PHC, so that programmes can be designed to support them. We suggest that there is a role for conducting community-based participatory research with FCHVs, involving methods that build the capacity of FCHVs to gather and share their experiences, including with policymakers, in ways that they want and value. While this kind of research can be resource-intensive, it has direct benefits to those involved, and could support the co-creation of policy relating to the delivery of UHC that reflects women volunteers' needs [65]. At the same time, policymakers and programme managers of FCHV programmes should consider arranging adequate monetary compensation for volunteers, not only to reflect the work context and the time spent in volunteering, but also as a livelihood strategy to support some of the poorest, rural women. This should be combined with activities to ease organisational bureaucracy and enhance community awareness of volunteers' roles, so that the FCHVs remain motivated to deliver towards UHC.

## Conclusion

Our paper illustrates how women from some of the poorest backgrounds can be highly motivated to work as a community health volunteer, yet inadequate family, community and health system support undermine this, the security of their livelihoods, and thus wider efforts to achieve UHC at primary care level for all. We found that insufficient payment, social and financial opportunity costs of volunteering, and out of pocket expenditure undermine motivation to deliver services. Similarly, the bureaucratisation of volunteers' work and a lack of social appreciation of their work by community members appeared to undermine volunteers' motivation. Financial investment to provide community volunteers with monetary compensation for their health work seems crucial if women in resource-poor areas of LMICs are to remain motivated to deliver PHC to achieve UHC for all; with this also being an investment in women's livelihoods and addressing gendered inequality.

## Supporting information

**S1 Checklist. Inclusivity in global research.**
(DOCX)

**S1 File. Interviews and focus groups guide.**
(DOCX)

## Acknowledgments

We would like to thank all the participants for their valuable time and sharing their experiences with us. We are grateful to Dr Rosie Steege (Liverpool School of tropical Medicine), Professsor Simon Rushton (the University of Sheffield) and Professor Ewen Speed from the University of Essex for revision of the paper to improve the manuscript. We would also like to

acknowledge Professor Paul Bissel and Professor Padam Simkhada for supervision while writing the main research report on which the paper is based.

## Author Contributions

**Conceptualization:** Sarita Panday, Edwin van Teijlingen, Amy Barnes.

**Data curation:** Sarita Panday, Edwin van Teijlingen, Amy Barnes.

**Formal analysis:** Sarita Panday, Amy Barnes.

**Methodology:** Edwin van Teijlingen.

**Supervision:** Edwin van Teijlingen.

**Writing – original draft:** Sarita Panday, Edwin van Teijlingen, Amy Barnes.

**Writing – review & editing:** Sarita Panday, Edwin van Teijlingen, Amy Barnes.

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
