## [Decision Letter · Decision Letter 0]

5 Mar 2024

PGPH-D-23-02589

Exploring the motivations of female community health volunteers in primary healthcare provision in rural Nepal: a qualitative study

Dear Dr. Panday,

Thank you for submitting your manuscript to PLOS Global Public Health. After careful consideration, we feel that it has merit but does not fully meet PLOS Global Public Health’s publication criteria as it currently stands. Therefore, we invite you to submit a revised version of the manuscript that addresses the points raised during the review process.

**Please submit your revised manuscript by March 31st, 2024. **If you will need more time than this to complete your revisions, please reply to this message or contact the journal office at globalpubhealth@plos.org. Please include the following items when submitting your revised manuscript:

We look forward to receiving your revised manuscript.

Kind regards,

Ziyue Wang, MBBS

Guest Editor

Journal Requirements:

Additional Editor Comments (if provided):

Reviewers' comments:

Reviewer's Responses to Questions

**Comments to the Author**

1. Does this manuscript meet PLOS Global Public Health’s publication criteria? Is the manuscript technically sound, and do the data support the conclusions? The manuscript must describe methodologically and ethically rigorous research with conclusions that are appropriately drawn based on the data presented.

Reviewer #1: Yes

Reviewer #2: Partly

Reviewer #3: Yes

2. Has the statistical analysis been performed appropriately and rigorously?

Reviewer #1: N/A

Reviewer #2: N/A

Reviewer #3: N/A

3. Have the authors made all data underlying the findings in their manuscript fully available (please refer to the Data Availability Statement at the start of the manuscript PDF file)?

Reviewer #1: Yes

Reviewer #2: Yes

Reviewer #3: Yes

4. Is the manuscript presented in an intelligible fashion and written in standard English?

Reviewer #1: Yes

Reviewer #2: Yes

Reviewer #3: Yes

5. Review Comments to the Author

Reviewer #1: This paper examines the factors influencing the motivations of Female Community Health Volunteers (FCHVs) in rural Nepal, which reveal truly interesting and significant issues regarding primary health care delivery. Here are several points that the authors can improve before publication:

1. The authors mentioned gender inequality and the relevant hierarchy. How does gender inequality or discrimination influence the FCHVs' motivation? Is it also a factor of motivation (or disincentives)?

2. The authors analyzed the qualitative data using thematic analysis and generated themes at individual, organizational, and community levels, it would be more helpful and clearer to understand if the authors could illustrate and summarize the different themes/subthemes using a structural figure or framework in the result part.

3. The data was collected in 2014, whereas the authors mentioned the FCHV programme was revised in 2010 and 2019, I wonder whether there can be any factors that may alter the motivation so far due to the policy revision. Despite the authors mentioning the number of FCHVs and their roles remain similar, a brief summary of the policy content change (or what does not change) would be helpful to justify the potential application of results to current circumstances.

4. The questionnaire of semi-structured interview and the list of focus group questions are recommended to be attached in the supplementary appendix.

5. It might be out of the scope of this study, but I am interested in whether the qualitative results are systematically different between the two selected regions, as the authors emphasized that the two rural settings were selected based on different characteristics.

Minor:

1. The numbering of the result part is messed up, and the format of headings and main texts is not consistent (e.g. main texts before and after 4.2 are in different line spacing), please correct them.

2. For the organizational-level and community-level factors, only one factor was analyzed in detail (Bureaucratisation and community perception), the extra subtitles seem redundant.

Reviewer #2: Thanks for inviting me to review this research article “Exploring the motivations of female community health volunteers in primary healthcare provision in rural Nepal: a qualitative study”. Based on the qualitative investigation and analysis, FCHVs’ motivations to volunteer in rural Nepal were affected in several unique ways. After evaluating this study, I recommend it for publication in this Journal, with revisions. Below are my comments and suggestions.

1. In the method section, authors need to describe the settings the way such as the meeting rooms/open spaces where the interviews/focus groups were conducted.

2. In the method section, compared to the volunteer participation, can authors provide more detail of the paid participation in this study such as the amount of payment or covered costs?

3. The authors present the socio-demographic characteristics of FCHVs (Table 2), age is also an important factor but it hasn’t been reported. For any unreported data in the tables, please indicate it as unknown.

4. In every narrative extract, can authors briefly describe the narrators (e.g. socio-demographics) to facilitate reading?

5. In the discussion section, more discussion of potential integrated solutions across three levels (individual, organization, and community) would be sound.

Reviewer #3: This paper by Panday et al. was a qualitative study exploring the motivations of female CHWs who are volunteers in delivering PHC services in rural Nepal. The authors conducted 31 semi-structured interviews and 3 focus groups among female CHWs including volunteers and paid workers in two districts of Nepal. Data was analyzed thematically and organized using the conceptual framework developed by Franco et al. Individual, organizational, and community level results included: 1) volunteers’ commitment to supporting mothers and children; disincentives and opportunity costs of volunteering; 2) bureaucratisation of volunteer work; and 3) community perception of volunteers. The authors concluded more support is needed at all these levels, and financial investments are necessary.

The paper addresses a clear research gap. Strengths include a straightforward background section and discussion which highlighted the relevance of the study. The conclusions drawn are supported by the results, with minor areas for improvement, and the method and discussion sections need some more detail. The main weakness is a results section presented without the use of tables or figures. This makes it challenging to synthesize the findings. There are also several typographical errors requiring copyediting. Overall, I recommend this paper is accepted with revisions.

Major areas of improvement include:

1) The lack of tables or figures in the results section. I strongly suggest creating a table or figure which organizes the results by individual, organizational, and community levels. Within each of these levels the authors can include their subthemes and draw connections between them. An example is provided by Figure 1 in a paper on the contribution of FCHVs to maternity care in Nepal also by Panday et al.

Panday S, Bissell P, Van Teijlingen E, Simkhada P. The contribution of female community health volunteers (FCHVs) to maternity care in Nepal: a qualitative study. BMC health services research. 2017 Dec;17(1):1-1.

2) I suggest the authors summarize their three tables on participant information into just one table.

3) In the introduction the authors describe data was collected for this study nine years ago. They do a good job defending the continued relevance of their findings, but maybe they can include a line on how this paper makes a different contribution from their earlier paper, which draws from the same data.

4) Some more detail is needed in the reporting of the thematic analysis in the methods section. For example, Braun and Clarke suggest creating initial thematic maps. Perhaps a codebook could be included as an appendix too.

5) The discussion section occasionally needs greater detail when connecting with the broader literature. For example, on page 23 the authors write “These findings are consistent with CHW literature from sub-Saharan Africa and South Asia (51, 52)”. Please expand what are the findings from these other studies if relevant.

6) Since the authors conclude more financial investment is needed to support FCHVs, I strongly suggest including more extracts explicitly identifying this (presently only extracts 5, 6, 8, 9, 16 mention salary/finances).

Copyediting is needed for this paper. Minor areas of improvement include:

1) Consistency between using the words “female” or “women”.

2) Numbers nine and below should be spelled out.

3) The format the date is presented in line 213 may be ambiguous.

4) Replace the apostrophe symbol in the tables.

5) Use oxford commas.

6) Fix punctuation such as double periods and spaces.

My only other comment is the authors write “informed consent was obtained from most of the participants”. I am confused how data was collected then for participants who did not provide consent.

I appreciate the opportunity to review this paper and thank the authors for their empirical contribution to improving PHC services in Nepal. I am available for re-review and questions from the editor.

6. PLOS authors have the option to publish the peer review history of their article (what does this mean?). If published, this will include your full peer review and any attached files.

**Do you want your identity to be public for this peer review?** For information about this choice, including consent withdrawal, please see our Privacy Policy.

Reviewer #1: No

Reviewer #2: No

Reviewer #3: No

---

## [Decision Letter · Decision Letter 1]

11 Jun 2024

Exploring the motivations of female community health volunteers in primary healthcare provision in rural Nepal: a qualitative study

PGPH-D-23-02589R1

Dear Dr Panday,

We are pleased to inform you that your manuscript 'Exploring the motivations of female community health volunteers in primary healthcare provision in rural Nepal: a qualitative study' has been provisionally accepted for publication in PLOS Global Public Health.

Best regards,

Sarang Deo, PhD

Academic Editor

All the reviewers, who also reviewed the earlier version of the manuscript, are satisfied with the author responses and revisions made to the manuscript. They unanimously recommend accepting the paper. There were no additional comments from the previous associate editor. Since I joined the review process this round, it's not fair for me to raise new set of concerns or comments. Hence, I recommend that the manuscript be accepted for publication.

Reviewer Comments (if any, and for reference):

Reviewer's Responses to Questions

**Comments to the Author**

1. If the authors have adequately addressed your comments raised in a previous round of review and you feel that this manuscript is now acceptable for publication, you may indicate that here to bypass the “Comments to the Author” section, enter your conflict of interest statement in the “Confidential to Editor” section, and submit your "Accept" recommendation.

Reviewer #1: All comments have been addressed

Reviewer #2: All comments have been addressed

Reviewer #3: All comments have been addressed

2. Does this manuscript meet PLOS Global Public Health’s publication criteria? Is the manuscript technically sound, and do the data support the conclusions? The manuscript must describe methodologically and ethically rigorous research with conclusions that are appropriately drawn based on the data presented.

Reviewer #1: Yes

Reviewer #2: Yes

Reviewer #3: Yes

3. Has the statistical analysis been performed appropriately and rigorously?

Reviewer #1: Yes

Reviewer #2: Yes

Reviewer #3: N/A

4. Have the authors made all data underlying the findings in their manuscript fully available (please refer to the Data Availability Statement at the start of the manuscript PDF file)?

Reviewer #1: No

Reviewer #2: Yes

Reviewer #3: Yes

5. Is the manuscript presented in an intelligible fashion and written in standard English?

Reviewer #1: Yes

Reviewer #2: Yes

Reviewer #3: Yes

6. Review Comments to the Author

Reviewer #1: The authors have made substantial improvement in the manuscript. Overall, it makes a significant contribution to the field. Well done!

Reviewer #2: Thanks for the invitation. I don't have additional comments for the authors.

Reviewer #3: Thank you for addressing my comments.

7. PLOS authors have the option to publish the peer review history of their article (what does this mean?). If published, this will include your full peer review and any attached files.

**Do you want your identity to be public for this peer review?** For information about this choice, including consent withdrawal, please see our Privacy Policy.

Reviewer #1: No

Reviewer #2: No

Reviewer #3: No
